# Content2Vec: Specializing Joint Representations of Product Images and Text for the task of Product Recommendation

**Thomas Nedelec, Elena Smirnova & Flavian Vasile**
Criteo Research
Paris, 32 Blanche, France
`{t.nedelec,e.smirnova,f.vasile}@criteo.com`

## Abstract

We propose a unified product embedded representation that is optimized for the task of retrieval-based product recommendation. We generate this representation using Content2Vec, a new deep architecture that merges product content information such as text and image, and we analyze its performance on hard recommendation setups such as cold-start and cross-category recommendations. In the case of a normal recommendation regime where collaborative information signal is available, we merge the product co-occurrence information and propose a second architecture Content2vec+ and show its lift in performance versus non-hybrid approaches in both cold start and normal recommendation regimes.

## 1 Introduction

Online product recommendation is now a key driver of demand, not only in E-commerce businesses that recommend physical products, such as Amazon (Marshall, 2006), TaoBao (Xiang, 2013) and Ebay (Academy, 2013), but also in online websites that recommend digital content such as news (Yahoo! - Agarwal et al. (2013), Google - Liu et al. (2010)), movies (Netflix - Bell & Koren (2007)), music (Spotify - Johnson (2015)), videos (YouTube - Covington et al. (2016)) and games (Xbox - Koenigstein et al. (2012)).

Two of the most challenging aspects of recommendation in general and of product recommendation in particular, are scalability and freshness. The first one addresses the problem of making fast recommendations in parallel, the second addresses the problem of updating recommendations based on real-time user interaction. One of the most encountered architecture solutions for recommendation at scale divides the recommendation process in two stages: *a candidate generation stage* that prunes the number of recommendable items from billions to a couple of hundreds, followed by a second *item selection stage* that decides the final set of items to be displayed to the user, as shown in Figure 1 (see Mazare (2016), Cheng et al. (2016), Covington et al. (2016)).

The first stage generally implies the pre-generation of an inverted index over the set of recommendable products, paired with a real-time retrieval module, similarly to a search engine architecture. In our current paper we focus on the cases where the system supports vectorial product queries. The sources of the vectorial representations range from the set of co-occurring products, like in the case of neighborhood-based collaborative filtering, to a low-dimensional representation produced via matrix factorization or to an embedded representation produced via a deep neural network.

The second stage takes the candidate set and decides the final list of recommendations, usually by optimizing a ranking metric. This stage has in general a lot more constraints in terms of latency, due to its use of real-time signal that makes its predictions not cacheable. Therefore, in terms of model choice, the first stage can be a lot more complex than the second. In terms of impact, the quality of the candidate set coming from the first stage is crucial, since this constitutes a hard threshold on the performance of the second stage and of the overall system.

Because of the feasibility of using a more complex model and the potential impact on the final recommendation performance, we choose to concentrate our efforts on the task of optimal candi-

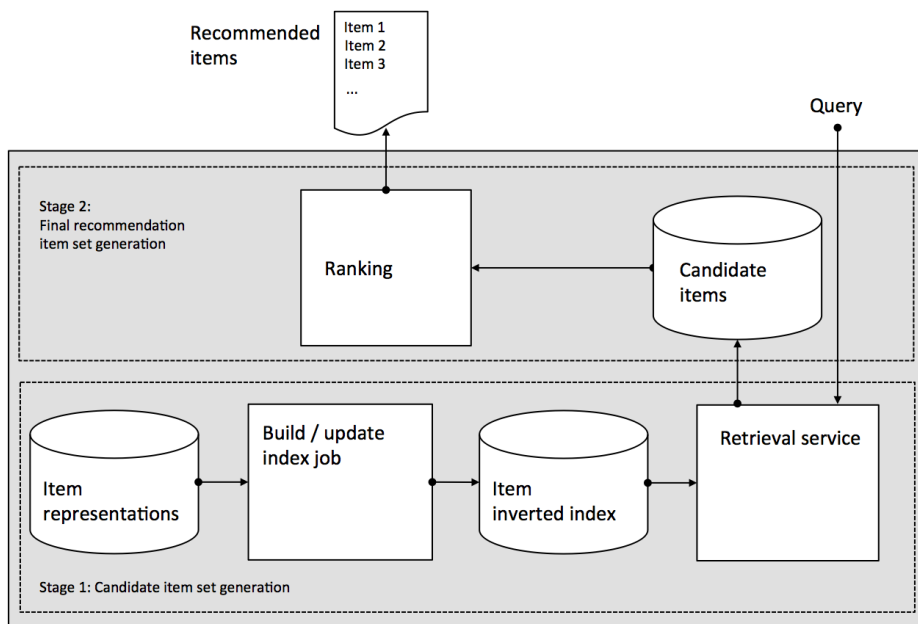

Figure 1: 2-Stage Recommender System Architecture.

date generation. We formalize the problem as a link prediction task, where given a set of past co-purchased products we try to predict unseen pairs of products. Related work in representation learning for recommendation investigated the use of collaborative filtering (CF), text and product images, but to our knowledge, there has been no attempt to unify all of these signals in a single representation. We see this as an opportunity to investigate the leveraging effect of generating a *Unified Product Representation* via a deep-learning approach. In the following, we formally define the set of associated requirements we would like to satisfy:

- **Relevance**: the representation should be optimized for product recommendation relevance, as measured by the associated target metrics (in this case, modeling it as a link prediction task and optimizing for the AUC of product pair prediction).

- **Coverage**: the representation should leverage all available product information (in our case, all product information available in the product catalog together with observed product co-occurrences).

- **Cross-modality expressiveness**: the representation should be able to account for interactions between various information sources such as text and image (can take into account the fact that the word "red" and the "red" color detector are correlated).

- **Pair-wise expressiveness**: the representation should be able to account for interactions between the two products.

- **Robustness**: the representation should operate well (recommendation performance will not degrade dramatically) in hard recommendation situations such as product cold-start (new products, new product pairs) and cross-category recommendation. These are important use-cases in product recommendation, when the product catalog has high churn (as in the case of flash sales websites or classifieds) or the recommendation needs to leverage cross-advertiser signal (as in the case of new users and user acquisition advertising campaigns). This is a different goal from simply trying to optimize for relevance metrics, due to the inherent limitations of offline metrics in predicting future online performance.

- **Retrieval-optimized**: the representation should be adapted to a content-retrieval setup, both on the query and on the indexing side, meaning that the vectors should be either small, sparse or both.

We propose a modular deep architecture that leverages state-of-the-art architectures for generating embedded representations for image, text and CF input, re-specializes the resulting product embeddings and combines them into a single product vector. This is a very general architecture that can plugin any networks in the image and text domain and re-use them for the problem of product recommendation, along with their gains in representation learning for the two domains. We investigate multiple ways of merging the modality-specific product information and propose a new type of residual-inspired unit, which we name *Pairwise Residual Unit*, that can model the joint aspects of the different product embeddings and show that it leads to good improvements.

We analyze our proposed architecture on an Amazon dataset (McAuley et al., 2015) containing information on co-purchased products. We report our improvements versus a text and an image-based baseline, that was introduced in previous work by (cite Julian) and show improvements both on normal and hard recommendation regimes such as cold-start and cross-category setups.

Our approach is similar to the recent work by (Covington et al., 2016), that propose a solution for video recommendation at YouTube. Unlike their proposed solution, where, in order to support user vector queries, the candidate generation step co-embeds users and items, we are interested to co-embed just the product pairs, which generally has a much smaller dimension. In our approach, the personalization step can happen after the per-item candidates are retrieved.

Our main contributions are the following:

- We propose a novel way of integrating deep-learning item representation in the context of large scale recommender system with a 2-stage serving architecture and introduce the new task of *Unified Product Representation* for optimal candidate selection in both cold start and normal recommendation setups.
- We introduce a new deep architecture that merges content and CF signal for the task of product recommendation and propose the *Pairwise Residual Unit*, a new learning component that models the joint product representations.
- We introduce two novel experimental setups (hard cold start, cross-category) and test that the proposed *Content2Vec* architecture satisfies the requirements we defined.

Though the focus of our work is on improving product recommendation through representation learning, we believe that simple extensions of our approach can be applied to many other recommendation scenarios.

The rest of the paper goes as follows: In Section 2 we cover previous related work and the relationship with our method. In Section 3 we present the Content2Vec model, followed by a detailed description of the resulting architecture in Section 4. In Section 5 we present the experimental setup and go over the results on Section 5.2. In Section 6 we summarize our findings and conclude with future directions of research.

## 2 RELATED WORK

Our work fits in the new wave of deep learning based recommendation solutions, that similarly to classical approaches can fall into 3 categories, namely collaborative filtering based, content based or hybrid approaches.

Several approaches use neural networks to build better item representations based on the co-occurrence matrix. The Prod2Vec algorithm (see (Grbovic et al., 2015)) implements Word2Vec ((Mikolov et al., 2013a), (Shazeer et al., 2016)), an algorithm that is at origin a shallow neural language model, on sequences of product ids, to reach a low-dimensional representation of each product. Among other embedding solutions that use the item relationship graph are the more recent extensions to Word2Vec algorithm such as Glove (Pennington et al., 2014) and SWIVEL (Shazeer et al., 2016) and the graph embedding solutions proposed in Node2Vec (Grover & Leskovec, 2016) and SDNE (Wang et al., 2016).

Content-based methods recommend an item to a user based upon an item description and a user profile ((Pazzani & Billsus, 2007)). This idea was deeply investigated in the information retrieval literature: in the context of web search, DSSM (Huang et al., 2013) and its extensions (Shen et al., 2014)(C-DSSM) and (Shan et al., 2016) are some of the most successful methods that specialize

query and document text embedding in order to predict implicit feedback signal such as document click-through rate. In the context of product recommendation, in (McAuley et al., 2015) the authors feed a pre-trained CNN (CNN trained on the ImageNet dataset, which is an image classification task that is very different from the task of image-based product recommendation) with products images and use the last layer of the network as the product embedding. This representation is subsequently used to compute similarities between products. Similarly, the authors in (Van den Oord et al., 2013) use CNNs to compute similarities between songs. Yosinski et al. (2014) show that the low layers of DNNs trained on different tasks are often similar and that good performance can be reached by fine-tuning a network previously trained on another task. In the case of recommendation systems, this fine tuning was implemented in Veit et al. (2015), where the authors specialize a GoogLeNet architecture to the task of predicting co-view events based on product pictures.

The performance of Collaborative Filtering (CF) models is often higher than that of content-based ones but it suffers from the cold-start problem. To take advantage of the best of both worlds, hybrid models use both sources of information in order to make recommendations. One possible way to incorporate product information is using it as side information in the product sequence model, as proposed in Meta-Prod2Vec (Vasile et al., 2016), leading to better product embeddings for products with low signal (low number of co-occurrences). In this work we continue the investigation of using both types of signal, this time both at training and product recommendation time.

## 3 CONTENT2VEC MODEL

Our proposed approach takes the idea of specializing the input representations to the recommendation task and generalizes it for multi-modality inputs, in order to leverage all product information and in particular, product images and product title and description text.

The main criteria for the Content2Vec architecture is to allow us to easily plugin new sources of signal and to replace existing embedding solutions with new versions. We are also interested in separating product-level embeddings from pair-level embeddings, such that the network can generate product vectors that are readily indexable. As a result, the Content2Vec architecture has three types of modules, as shown in Figure 2:

- **Content-specific embedding modules** that take raw product information and generate the associated vectors. In this paper we cover embedding modules for text, image, categorical attributes and product co-occurrences (for an example, see Figure 3).
- **Overall product embedding modules** that merge all the product information into a unified product representation.
- **Pair embedding module** that merges the product-to-product interactions and computes the final similarity score. In the case of retrieval-optimized product embeddings, this module becomes the inner-product between the two items and all interactions between them are to be approximated within the product-level embedding modules.

Content2Vec training follows the architecture, learning module-by-module. In the first stage, we initialize the content-specific modules with embeddings from proxy tasks (classification for image, language modeling for text) and re-specialize them to the task of product recommendation. For the specialization task, as mentioned in Section 1, we frame the objective as a link prediction task where we try to predict the pairs of products purchased together. We describe the loss function in Section 3.1.

In the second stage, we stack the modality-specific embeddings generated in the first stage into a general product vector and learn an additional residual vector using the same learning objective as in the specialization step. This will described in depth in Section 4.2.

Finally, in the third stage, given the updated product vectors from stage two, we learn the linear combination between the similarities of the product vectors and make the final prediction.

### 3.1 LOSS FUNCTION

The previous work on learning pair-wise item distances concentrated on using ranking (McFee & Lanckriet, 2010), siamese (Hadsell et al., 2006) or logistic loss (Zheng et al., 2015). For optimizing the link prediction objective we choose the logistic similarity loss (eq. 1) that has the advantage of

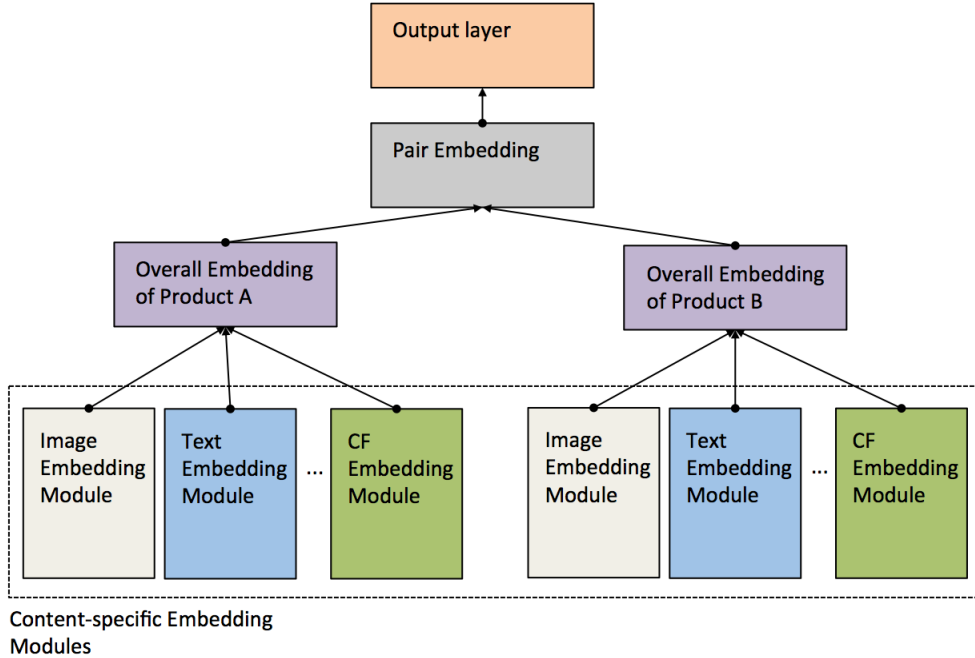

Figure 2: Content2Vec architecture combines content-specific modules with residual vector to produce embedding vector for each product, then uses these vectors to compute similarities between products.

having a fast approximation via Negative Sampling loss (Mikolov et al., 2013b) shown in eq. 2. By using Negative Sampling, the prediction step can scale up to large number of items, by using all positive pairs and sampling the negatives on the fly.

$$L(\theta) = \sum_{ij} -X_{ij}^{POS} \log \sigma(sim(a_i, b_j)) - X_{ij}^{NEG} \log \sigma(-sim(a_i, b_j)) \tag{1}$$

$$L_{NS}(\theta) = \sum_{ij} -X_{ij}^{POS} (\log \sigma(sim(a_i, b_j)) + \sum_{l=1}^{k} \mathbb{E}_{n_l \sim P_D} \log \sigma(-sim(a_i, n_l))) \tag{2}$$

where:
$\theta = (a_i, b_j)$ is the set of model parameters, where $a_i$ and $b_j$ are the embedding vectors for the products A and B,
$sim(a_i, b_j) = \alpha < a_i, b_j > +\beta$ is the similarity function between $a_i$ and $b_j$, $\alpha$ and $\beta$ are scalar values,
$X_{ij}^{POS}$ is the frequency of the observed item pair $ij$ (e.g. the frequency of the positive pair $ij$),
$X_{ij}^{NEG} = X_i - X_{ij}^{POS}$ is the frequency of the unobserved item pair $ij$ (we assume that all unobserved pairs are negatives),
$P_D$ probability distribution used to sample negative context examples $n_l$,
$k$ is a hyper parameter specifying the number of negative examples per positive example.

## 4 CONTENT2VEC MODULES

### 4.1 CONTENT-SPECIFIC EMBEDDING MODULES

Content-specific modules can have various architectures and are meant to be used separately in order to increase modularity. Their role is to map all types of item signal into embedded representations.

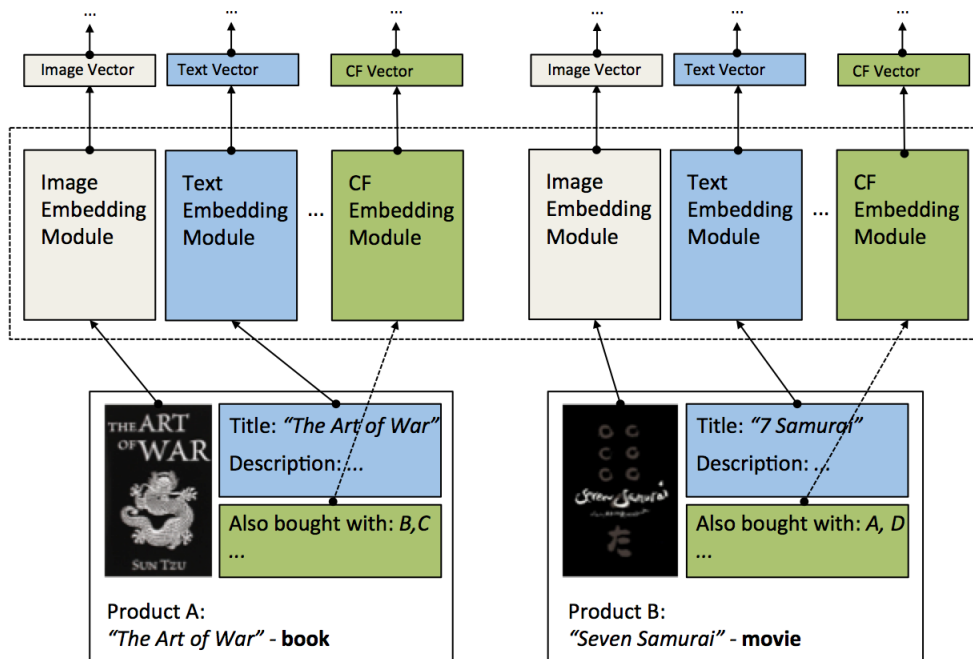

Figure 3: An example of using the content-specific modules to create embedded representations of two products with images, text and CF signal.

In Figure 3 we give an illustrative example of mapping a pair of products to their vectorial representations.

In the following we analyze four types of input signal and embedding solutions for each one of them. For all of the modules, we use $L_{NS}$ loss (see eq. 2) as specialization loss.

### 4.1.1 EMBEDDING PRODUCT IMAGES: ALEXNET

**Model and proxy task: CNN for Image Classification** For generating the image embeddings we propose reusing a model trained for image classification, as in previous work by (Krizhevsky et al., 2012) and (He & McAuley, 2015). In (He & McAuley, 2015), the authors have shown how to use the Inception architecture (Szegedy et al., 2015) and specialize it for the product recommendation task. However, the Inception architecture is very deep and requires extensive training time. For ease of experimentation we use AlexNet, which is a simpler architecture that was also a winner on the ImageNet task (Krizhevsky et al., 2012) previously to Inception NN. In section 5.2 we will show that, even if simpler, when combined with additional product text information, the AlexNet-based solution can perform very well on the recommendation task.

For our experiments, we use the pretrained version of AlexNet available on Toronto's university website. We experimented with two different ways to specialize the representation in order to compute product similarities. In the first one, we learn a weighted inner product between the two representations (fc7 layer of ImageNet). In the second one, we specialize the fc7 layer to detect product similarities. The second approach lead to much better performance and is the one for which we report final results.

### 4.1.2 EMBEDDING PRODUCT TEXT: WORD2VEC AND CNN ON SENTENCES

**Model and proxy task: Word2Vec for Product Language Modeling** For generating word embeddings, we propose reusing Word2Vec Mikolov et al. (2013b), a model for generating language models that has been employed in a various of text understanding tasks. More recently, it has been shown in (Pennington et al., 2014) that Word2Vec is closely linked with matrix factorization techniques applied on the word co-occurrence matrix. For Content2Vec, we chose to pretrain Word2Vec

on the entire product catalog text information and not use an available set of word embeddings such as the one created on the Google Corpus. The main reason is that the text distribution within product descriptions is quite different from the general distribution. For example the word *'jersey'* has a very different conditional distribution within the product description corpus versus general online text.

Text CNN (Kim, 2014) offers a simple solution for sentence-level embeddings using convolutions. The convolutions act as a form of n-gram filters, allowing the network to embed sentence-level information and specializing word embeddings to higher-order tasks such as text classification or sentiment analysis. To the best of our knowledge, this is the first attempt to employ them for the task of product recommendation. For our task, we generate sentences based on the product titles and descriptions.

### 4.1.3 EMBEDDING PRODUCT CO-OCCURRENCES: PROD2VEC

Prod2Vec (Grbovic et al., 2015) is an extension of the Word2Vec algorithm to product shopping sequences. As a result, Prod2Vec can be seen as a matrix factorization technique on the product co-occurence matrix. In Content2Vec, the Prod2Vec-based similarity contains all of the information that can be derived from the sequential aspect of the user behavior, without taking into account the per-product meta-data.

### 4.1.4 EMBEDDING CATEGORICAL PRODUCT META-DATA: META-PROD2VEC

Meta-Prod2Vec (Vasile et al., 2016) improves upon Prod2Vec by using the product meta-data side information to regularize the final product embeddings. In Content2Vec, we can use the similar technique of co-embedding product categorical information with product ids to generate the embedding values for the categorical features.

### 4.2 JOINT PRODUCT EMBEDDING: PAIRWISE RESIDUAL UNIT

As stated in Section 1, the function of the product embedding module is two-fold: first, to model all interactions that exist between the modality-specific embeddings with respect to the final optimization objective, and second, to approximate interaction terms between the products that cannot be explained by a linear combination of the modality-specific similarities. With this in mind, we introduce a new type of learning unit, the *Pairwise Residual Unit* (eq. 4), which similarly to the original *residual unit* introduced in He et al. (2015) (eq. 3), allows the layers to learn incremental, i.e. residual representations (see Figure 4).

In Hardt & Ma (2016) the authors motivate the use of residual units as helping preserve the representations learned in the previous layers. In our case we are interested in preserving the specialized image and text representations and learn an additional representation for their interactions. Though in previous work, most the of the residual units are using at least two ReLU layers in the residual unit, we observe good results using just one. In order to model interactions between modalities, we could also learn a fully connected layer initialized with identity that takes as input the concatenated modality-specific vectors. However, in order to have a smaller number of parameters and increase model comprehensibility, we would like to keep separate the modality-specific representations and to model the final prediction model as an ensemble.

$$y = F(x) + x \qquad (3)$$

$$y = sim(F(x_1), F(x_2)) + sim(x_1, x_2) \qquad (4)$$

where:
$x_1$ and $x_2$ are the two product embedding vectors (obtained by stacking the modality-specific vectors),
$sim(.,.)$ is a similarity function over two embedding vectors $x_1$, $x_2$,
$F(x)$ is a Rectified Linear Unit.

To be able to measure the incremental value of introducing a residual vector we introduce a baseline architecture that computes the final prediction based on the linear combination of the modality-specific similarities denoted by *Content2Vec-linear* with the associated similarity function defined in eq. 5.

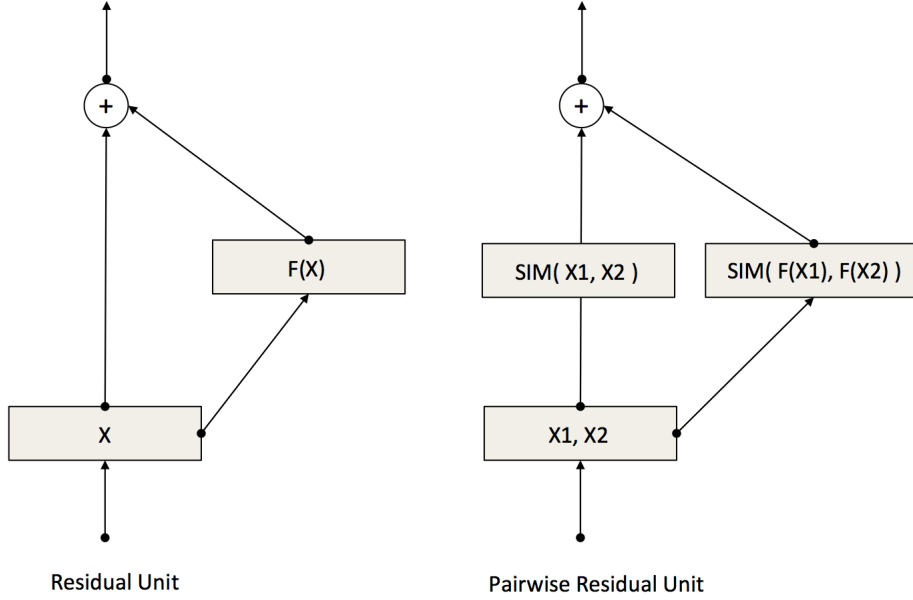

Figure 4: Pairwise Residual Unit

$$sim_{c2v}(a_i, b_j) = \sum_{m \in Modalities} w_m \sigma(sim_m(a_i, b_j)) \qquad (5)$$

Under this notation, the residual-based architecture denoted as *Content2Vec-res* minimizes $L_{NS}$ with the similarity function defined in eq. 6.

$$sim_{c2v-res}(a_i, b_j) = \sum_{m \in (Modalities+Residual)} w_m \sigma(sim_m(a_i, b_j)) \qquad (6)$$

In order to learn the residual vector, we keep fixed the modality-specific similarities and co-train the final weights of each of the modalities together with the product-specific residual layers. For example, in the case of using only image and text signals, our final predictor can be defined as in eq 7, where $P_{txt}$ and $P_{img}$ are pre-set and $w_{txt}$, $w_{img}$, $w_{res}$ and $P_{res}$ are learned together:

$$P(pos|a,b) = \sigma(w_{txt}P_{txt}(pos|a_{txt}, b_{txt}) + w_{img}P_{img}(pos|a_{img}, b_{img}) + w_{res}P_{res}(pos|a_{res}, b_{res})) \quad (7)$$

where: *pos* is the positive outcome of products A and B being bought together and $P_{res}(pos|a,b) = \sigma(\alpha < F([a_{txt}, a_{img}]), F([b_{txt}, b_{img}]) > +\beta)$

In Section 5.2 we compare the performance of *Content2Vec-res* and *Content2Vec-linear* and show that, as expected, the proposed architecture surpasses the performance of the linear model, while allowing for a retrieval-based candidate scoring solution.

## 4.3 PAIR EMBEDDING MODULE

In a retrieval-based architecture, the pair embedding module cannot support more than a simple linear combination of the product embedding vectors, such that the final score can be computed via inner-product. However, we are still interested to know the trade-off in performance between an inner-product-based candidate scoring and a model that allows for explicit interaction terms between the items. To this end, we introduce two explicit interaction models: *Content2Vec-crossfeat* - a

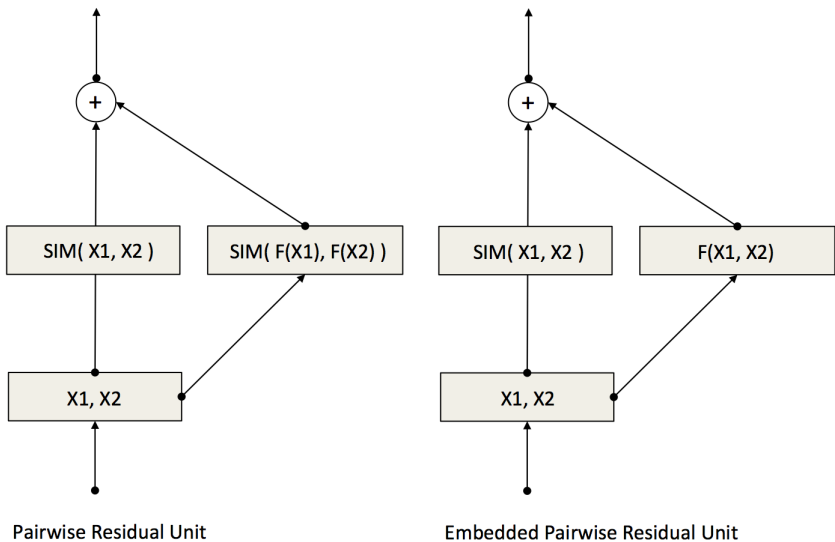

Figure 5: The two types of Pairwise Residual Units. By comparison with the first version that outputs a scalar, the second one outputs a vector that goes directly into the final prediction layer

model where we discretize the text and image-specific similarity scores and create explicit feature conjunctions between them and *Content2Vec-embedpairs* - a model where we use a similar technique with *Paiwise Residual Unit*, in this case modeling the residual of the linear similarity directly as a vector in the pair embedding layer, as shown in Figure 5. In Section 5.2 we show that two models have as expected better performance than the linear model and that the pair embedding is slightly better.

# 5 EXPERIMENTAL RESULTS

## 5.1 DATASET

We perform our evaluation on the publicly available Amazon dataset (McAuley et al., 2015) that represents a collection of products that were co-bought on the Amazon website. Each item has a rich description containing product image, text and category (any of the modalities can be missing). In terms of dimensionality, the dataset contains around 10M pairs of products. We concentrate on the subgraph of Book and Movie product pairs, because both categories are large and they have a reasonable sized intersection. This allows us to look at recommendation performance on cross-category pairs (to evaluate a model trained only on Book pairs on predicting Movie co-bought items) and mixed category pairs (to evaluate the models on Book-Movie product pairs).

Based on the full Book & Movies data we generate three datasets with different characteristics:
The first dataset simulates a **hard cold start regime**, where all product pairs used in validation and testing are over products unseen in training. This tests the hardest recommendation setup, where all testing data is new. We decided to bench all of our hyperparameters on this regime and use the best setup on all datasets, since tuning on the harder dataset ensures the best generalization error (results shown in Table 1).
The second dataset simulates a **non-cold start regime**, where the vast majority of the products in the test set are available at training time. The dataset is generated by taking the top 100k most connected products in the original dataset and keeping the links between them (results shown in Table 2).
The third dataset simulates a **soft cold start regime**, where some of the products in the test set are available at training time. The dataset is generated by taking the top 200k most connected products in the original dataset and sampling 10% of the links between them (results shown in Table 3).

**Hyper-parameters**   We fixed the sizes of embedding vectors for image CNN module to 4096 hidden units, for text CNN module to 256, for Prod2Vec module to 50, for residual representation to 128. For optimization we use an Adam algorithm and we manually set the initial learning rate based on the validation set performance. The batch sizes vary for different datasets. We train all the models until validation set performance stops increasing.

**Evaluation task**   We evaluate the recommendation methods on the product link prediction task, similar to (He & McAuley, 2015). We consider the observed product pairs as positive examples and all unknown pairs as negatives. We generate negative pairs according to the popularity of the products in the positive pairs (negative examples between popular products are more likely to be generated) with a positive to negative ratio of 1:2.

**Evaluation metrics**   For the link prediction task, we use the Area Under Curve (AUC) of the Precision/Recall curve as our evaluation metric.

**Competing methods**

- *ImageCNN*: prediction based on specialized image embeddings similarity
- *TextCNN*: prediction based on specialized text embeddings similarity
- *Content2Vec-linear*: prediction based on the linear combination of text and image similarities
- *Content2Vec-crossfeat*: prediction based on the linear combination of discretized image and text similarities and their conjuctions
- *Content2Vec-res*: prediction based on the linear combination of text and image similarities plus product-level residual vectors similarities
- *Content2Vec-embedpairs*: prediction based on the linear combination of text and image similarities and a pair-level residual component
- *Prod2Vec*: prediction based on the product vectors coming from the decomposition of the co-purchase matrix
- *Content2Vec+*: prediction based on the ensemble of Prod2Vec and Content2Vec models

## 5.2   RESULTS

The results on hard and soft cold start datasets (Tables 1, 3) show that our main proposed method *Content2Vec-res* can leverage the additional signal provided by each of the input modalities in a joint manner and leads to significant gains in AUC versus the one-signal baselines (ImageCNN, TextCNN) and their linear combination (Content2Vec-linear).
From the point of view of robustness, *Content2Vec-res* learns product representations that perform better than the baseline methods on out-of-sample recommendations such as cross-category pairs and mixed-category pairs (Table 1).
We observe that adding an additional layer that represents pair-level interactions does not lead to big improvements in either of the two models we investigated (Content2Vec-crossfeat,embedpairs), confirming that a product retrieval-based recommender system can achieve state-of-the-art results.
Finally, *Content2Vec-res+*, our proposed hybrid architecture that combines content and CF signal achieves better performance than the content and CF-only models, with bigger lifts in the case of the third dataset (Table 3) where the CF signal is weaker due to higher sparsity.

| Recommendation Model | Books | Movies | Mixed |
|---|---|---|---|
| **Models trained on Books dataset** | | | |
| Book ImageCNN specialized | 81% | 78% | 64% |
| Book TextCNN | 72% | 79% | 76% |
| Book Content2Vec-linear | 83% | 83% | 76% |
| Book Content2Vec-crossfeat | 86% | 83% | 83% |
| Book Content2Vec-res | 89% | 83% | 77% |
| Book Content2Vec-embedpairs | 90% | 82% | 77% |
| **Models trained on Movies dataset** | | | |
| Movie ImageCNN specialized | 59% | 92% | 60% |
| Movie TextCNN | 63% | 90% | 65% |
| Movie Content2Vec-linear | 64% | 94% | 65% |
| Movie Content2Vec-crossfeat | 62% | 94% | 63% |
| Movie Content2Vec-res | 60% | 95% | 66% |
| Movie Content2Vec-embedpairs | 64% | 94% | 65% |

Table 1: AUC results of image and text-based embeddings on hard cold-start dataset on Book, Movie and Mixed category test product pairs.

| Recommendation Model | Test |
|---|---|
| Content2Vec-linear | 84% |
| Content2Vec-res | 87% |
| Prod2Vec | 96% |
| Content2Vec-linear+ | 97% |
| Content2Vec-res+ | 97% |

Table 2: AUC results on non cold-start dataset.

| Recommendation Model | Test |
|---|---|
| ImageCNN | 80% |
| TextCNN | 78% |
| Content2vec-linear | 88% |
| Content2vec-res | 89% |
| Content2vec-embed_pairs | 90% |
| Prod2vec | 86% |
| Content2vec-linear+ | 89% |
| Content2vec-res+ | 92% |
| Content2vec-embed_pairs+ | 92% |

Table 3: AUC results on soft cold-start dataset.

## 6 CONCLUSIONS

This work has several key contributions. We show how to use all product signal for the task of product recommendation using a modular architecture that can leverage fast evolving solutions for each type of input modality. We define a set of requirements for evaluating the resulting product embeddings and show that our method leads to significant improvements over the single signal approaches on hard recommendation situations such as cold-start and cross-category evaluation. Finally, in order to model the joint aspects of the product embeddings we introduce a new type of learning unit, named *Pairwise Residual Unit* and show the resulting gains on a real product co-purchases dataset. In the current work we have addressed all but one of the desired requirements, namely generating retrieval-optimized embeddings. For the next steps, we want to pursue sparse and compressed product representations, in order to help the performance of the final product retrieval system.

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
