# Peer review of "CONTENT2VEC: SPECIALIZING JOINT REPRESENTATIONS OF PRODUCT IMAGES AND TEXT FOR THE TASK OF PRODUCT RECOMMENDATION"

_ICLR 2017 — rejected_

[Author Response · Thomas Nedelec · 10 Dec 2016]
**Answer to the different questions**

Thanks for your questions. We shall try to adress them in the following:

Question 1: There are many modules in the system that are trained separately. Furthemore, fine-tuning for e.g. the image model appears to only take place on the FC7 layer. Jointly training all the components of the model would presumably improve performance, perhaps considerably. Did you try this? If not, why not?

No we did not. We were primarily interested in knowing the additional performance one can gain by merging pretrained models and re-specializing them to the task. We agree that for the paper it would have been a worthy experiment but we did not get to it due to time constraints. We will try to add it in the next paper update.

Question 2: For the joint product embedding, it was stated that a fully connected hidden layer that mixes the individual embeddings wasn't used in order to have a smaller number of parameters and increase comprehensibility. But presumably it could also perform better than the chosen approach. Did you try it?

We made some attempts but the convergence speed was bad. It is also on the to-do list for the next paper update.

Question 3: Could you provide some more details on the way TextCNN was used? Was it used on full product descriptions, even when they consist of multiple sentences? Were titles and descriptions embedded separately?

Sure! We decided to not embed the titles and descriptions separately, but concatenate the two and keep the first ten words without stop words (with empty word padding for shorter sequences). This method can definitely be extended to the full description since text CNN are quite robust to the length of the sequence.

Question 4:  In the table of results at the end, I didn't see any results for a model trained on both the books and movies data and tested on both in the cold start scenario. Did you evaluate this case?

Yes, we did try and as expected the results were not great, since the current version of the model does not support category-specific embeddings. Therefore, the resulting overall distance between products is a compromise between the two category-specific distances. If interested, we can provide you the results we achieved.

Question 5: Earlier work on content-based recommendation focused on not just performance but also on the interpretability of the resulting models. Can you say anything about whether effective interpretations of the models latent dimensions can be gleaned when learning in this way?

We only explored the text interpretability by looking at the top activating word embeddings  for each of the text convolutions. What we saw is that most of the 100 unigram filters used in the convolution layer corresponds to a topic (could be clusters corresponding to a genre, a date or an author). It looks like while learning the right text representation for book pairwise similarity prediction the model discovers naturally occuring book topics. The paper was already long so we decide not to publish them, but if you are interested, we can provide them.

Question 6: Is Prod2Vec realistic in cold-start scenarios? Presumably co-purchase information wouldn't be available for cold-start products and thus an embedding couldn't be estimated.

The reason why we keep Prod2vec only in Content2vec + is because prod2vec only works when you have available collaborative filtering signal on at least some of the products in test. That is why prod2vec and content2vec+ algorithm results are only reported in the two datasets with some CF signal.

Question 7: I didn't understand why different sets of baselines seem to appear across the three tables, but maybe I missed some detail here.

We used the hard cold start dataset to do the extensive analysis of our architecture VS other available methods. We dropped crossfeat because we made the case in the hard cold start dataset.

Feel free to comment if you have any other questions.

Elena, Flavian & Thomas

[Official Review · AnonReviewer3 · rating 5 · confidence 3 · 15 Dec 2016]
**No Title**

The paper proposes a method to combine arbitrary content into recommender systems, such as images, text, etc. These various features have been previously used to improve recommender systems, though what's novel here is the contribution of a general-purpose framework to combine arbitrary feature types.

Positively, the idea of combining many heterogeneous feature types into RS is ambitious and fairly novel. Previous works have certainly sought to include various feature types to improve RSs, though combining different features types successfully is difficult.

Negatively, there are a few aspects of the paper that are a bit ad-hoc. In particular:
-- There are a lot of pieces here being "glued together" to build the system. Different parts are trained separately and then combined together using another learning stage. There's nothing wrong with doing things in this way (and indeed it's the most straightforward and likely to work approach), but it pushes the contribution more toward the "system building" direction as opposed to the "end-to-end learning" direction which is more the focus of this conference.
-- Further to the above, this makes it hard to say how easily the model would generalize to arbitrary feature types, say e.g. if I had audio or video features describing the item. To incorporate such features into the system would require a lot of implementation work, as opposed to being a system where I can just throw more features in and expect it to work.

The pre-review comments address some of these issues. Some of the responses aren't entirely convincing, e.g. it'd be better to have the same baselines across tables, rather than dropping some because "the case had already been made elsewhere".

Other than that, I like the effort to combine several different feature types in real recommender systems datasets. I'm not entirely sure how strong the baselines are, they seem more like ablation-style experiments rather than comparison against any state-of-the-art RS.

[Official Review · AnonReviewer1 · rating 3 · confidence 3 · 18 Dec 2016]
**No Title**

This paper proposes combining different modalities of product content (e.g. review text, images, co-purchase info ...etc) in order to learn one unified product representation for recommender systems. While the idea of combining multiple sources of information is indeed an effective approach for handling data sparsity in recommender systems, I have some reservations on the approach proposed in this paper:

1) Some modalities are not necessarily relevant for the recommendation task or item similarity. For example, cover images of books or movies (which are product types in the experiments of this paper) do not tell us much about their content. The paper should clearly motivate and show how different modalities contribute to the final task.

2) The connection between the proposed joint product embedding and residual networks is a bit awkward. The original residual layers are composed of adding the original input vector to the output of an MLP, i.e. several affine transformations followed by non-linearities. These layers allow training very deep neural networks (up to 1000 layers) as a result of easier gradient flow. In contrast, the pairwise residual unit of this paper adds the dot product of two item vectors to the dot product of the same vectors but after applying a simple non-linearity. The motivation of this architecture is not very obvious, and is not well motivated in the paper.

3) While it is a minor point, but the choice of the term embedding for the dot product of two items is not usual. Embeddings usually refer to vectors in R^n, and for specific entities. Here it refers to the final output, and renders the output layer in Figure 2 pointless.

Finally, I believe the paper can be improved by focusing more on motivating architectural choices, and being more concise in your description. The paper is currently very long (11 pages) and I strongly encourage you to shorten it.

[Official Review · AnonReviewer2 · rating 3 · confidence 3 · 19 Dec 2016]
**Interesting problem and good motivation, unconvincing solution architecture**

The problem of utilizing all available information (across modalities) about a product to learn a meaningful "joint" embedding is an interesting one, and certainly seems like it a promising direction for improving recommender systems, especially in the "cold start" scenario. I'm unaware of approaches combining as many modalities as proposed in this paper, so an effective solution could indeed be significant. However, there are many aspects of the proposed architecture that seem sub-optimal to me:

1. A major benefit of neural-network based systems is that the entire system can be trained end-to-end, jointly. The proposed approach sticks together largely pre-trained modules for different modalities... this can be justifiable when there is very little training data available on which to train jointly. With 10M product pairs, however, this doesn't seem to be the case for the Amazon dataset (although I haven't worked with this dataset myself so perhaps I'm missing something... either way it's not discussed at all in the paper). I consider the lack of a jointly fine-tuned model a major shortcoming of the proposed approach.

2. The discussion of "pairwise residual units" is confusing and not well-motivated. The residual formulation (if I understand it correctly) applies a ReLU layer to the concatenation of the modality specific embeddings, giving a new similarity (after dot products) that can be added to the similarity obtained from the concatenation directly. Why not just have an additional fully-connected layer that mixes the modality specific embeddings to form a final embedding (perhaps of lower dimensionality)? This should at least be presented as a baseline, if the pairwise residual unit is claimed as a contribution... I don't find the provided explanation convincing (in what way does the residual approach reduce parameter count?).

3. More minor: The choice of TextCNN for the text embedding vectors seems fine (although I wonder how an LSTM-based approach would perform)... However the details surrounding how it is used are obscured in the paper. In response to a question, the authors mention that it runs on the concatenation of the first 10 words of the title and product description. Especially for the description, this seems insufficiently long to contain a lot of information to me.

More care could be given to motivating the choices made in the paper. Finally, I'm not familiar with state of the art on this dataset... do the comparisons accurately reflect it? It seems only one competing technique is presented, with none on the more challenging cold-start scenarios.

Minor detail: In the second paragraph of page 3, there is a reference that just says (cite Julian).

[Author Response · Thomas Nedelec · 09 Jan 2017]
**Reply to reviews**

Dear reviewers, 
 
thanks for your detailed reviews. We will use them to improve our paper. We tried to gather them in different categories in oder to address them properly. 
Feel free to come back to us if you have some remaining questions.
 
Elena, Flavian & Thomas
 
Common themes:
     • We do not do end-to-end learning (I.1. & III.1)
        ◦ TL;DR: A modular architecture is the easiest way to put in production such a complex model - efficient way of computational resources (as also ack by the first reviewer)
           ▪ the fact that we specialize each modality-specific network to the final task independently allows us to scale better by reducing the parameter space and the resulting computation time, gives us better interpretability and insight into the relative value of each modality (as raised by reviewer #2) and allows us to better control any convergence issues in a production environment.
           ▪ the fact that during specialization we do not backprop through the full ImageCNN network is based on previous findings (see 'Learning visual clothing style with heterogeneous dyadic co-occurences’ for instance)  that show that the fc7 layer provides a good cross-task representation for images. For the other two networks we do backprop all the way down.
         ◦ Using pretrained models for each modality allows us to leverage external source of data and do transfer learning, whose value was repeatedly confirmed in the literature such as 'How transferable are features in deep NN?' for instance. 
         ◦ We agree that in the limit given enough data and computational resources the models can be learnt from scratch and in a joint manner, but our concern is on delivering a simple solution that can be delivered in production setting and at very large scale.
     • ResNet motivation (I.2. & II.2)
         ◦ We decided to use “residual” in describing our new pairwise embedding layer based on the similarity in motivation with the original residual unit that was introduced to help the system approximate the identity and allows the new layers to focus on the remaining unexplained error. We agree that in our case the layer does not serve the same practical purpose as in the original ResNets architectures that use the unit mostly for training very deep networks.
     • Missing baselines
         ◦ the state of the art on this dataset (I.3.b & III.3.)
            ▪ ImageCNN is the McAuley et al solution that introduced the dataset (He used GoogLeNet and we used AlexNet)
            ▪ VBPR is another baseline introduced by He & McAuley. It uses reviews as implicit feedback and try to predict the interest of a user based on products images and the sequence of products he was interested in. Our approach focuses more on a item to item similarity matrix and that is why we have not implemented this approach.   
         ◦ jointly fine-tuned model (I.1)
            ▪ We agree that we could respecialize the networks jointly and its on our list of remaining experiments to try 
         ◦ additional fully-connected layer that mixes the modality specific embeddings to form a final embedding (I.2)
            ▪ We agree that having an additional full connected layer that compress into a lower dimension vector could be a good idea. However we did not manage to make it work. We’ll try different initialisations in the next version of the paper.
 
Other concerns:
   • TextCNN setup details (I.3.a.)
         ◦ As per not providing more info on training the TextCNN, we choose to not add more info as our paper was running long as it was. For more specific info, please let us know what details are you interested in.
         ◦ Concerning the choice of keeping only the first 10 words, we agree that we could have kept more, however a lot of the descriptions were empty so we did not feel we need to make them longer. However, the architecture allows to choose an arbitrary length for the text input. 
    • The paper should clearly motivate and show how different modalities contribute to the final task. (II.1.)
         ◦ We agree and this is why we reported separately the performance of each modality network in Tables 1 & 2
         ◦ Something we did not mention is that in Content2Vec-linear model the image and text modalities have very similar coefficients 
    • The choice of the term embedding for the dot product of two items is not usual.  Here it refers to the final output, and renders the output layer in Figure 2 pointless. (II.3.a.)
         ◦ In the Content2Vec-crossfeat and Content2Vec-embedpairs we do have a pair embedding layer (one using polynomial features, one using a real embedding representation) so this is why we differentiate  between the Output layer (the softmax) and the Pairwise layer in Figure 2.
    • This architecture makes it hard to say how easily the model would generalize to arbitrary feature types, say e.g. if I had audio or video features describing the item. (III.2.)
         ◦ The modular design assumes that the input modalities do have extra modeling value for the pairwise similarity prediction. To your point, if interested in song similarities, if shown that text and audio signals do separately predict video similarities (as shown in

[Final Decision · Program Chairs · 06 Feb 2017]
**ICLR committee final decision**

The idea of combining many modalities for product recommendation is a good one and well worth exploring. However, the approach presented in this paper is unsatisfying, as it involves combining several pre-trained models, in a somewhat ad hoc manner. Overall a nice problem, but the formulation and results are not presented clearly enough.